# The Bioactive Properties of Carotenoids from Lipophilic *Sea buckthorn* Extract (*Hippophae rhamnoides* L.) in Breast Cancer Cell Lines

**DOI:** 10.3390/molecules28114486

**Published:** 2023-06-01

**Authors:** Simona Visan, Olga Soritau, Corina Tatomir, Oana Baldasici, Loredana Balacescu, Ovidiu Balacescu, Patricia Muntean, Cristina Gherasim, Adela Pintea

**Affiliations:** 1Department of Genetics, Genomics, and Experimental Pathology, The Oncology Institute “Prof. Dr. Ion Chiricuta”, 400015 Cluj-Napoca, Romania; oana_baldasici@yahoo.ro (O.B.); loredana_balacescu@yahoo.com (L.B.); obalacescu@yahoo.com (O.B.); 2Department of Cell Biology and Radiobiology, The Oncology Institute “Prof. Dr. Ion Chiricuta”, 400015 Cluj-Napoca, Romania; olgasoritau@yahoo.com (O.S.); coratat@yahoo.com (C.T.); 3Department of Chemistry and Biochemistry, University of Agricultural Sciences and Veterinary Medicine, 400372 Cluj-Napoca, Romania; patricia-andrea-lia.muntean@usamvcluj.ro (P.M.); cristina.gherasim@usamvcluj.ro (C.G.); apintea@usamvcluj.ro (A.P.)

**Keywords:** *Sea buckthorn*, carotenoids, cytotoxicity, apoptosis, breast cancer, antioxidants, pro-oxidants

## Abstract

In women, breast cancer is the most commonly diagnosed cancer (11.7% of total cases) and the leading cause of cancer death (6.9%) worldwide. Bioactive dietary components such as *Sea buckthorn* berries are known for their high carotenoid content, which has been shown to possess anti-cancer properties. Considering the limited number of studies investigating the bioactive properties of carotenoids in breast cancer, the aim of this study was to investigate the antiproliferative, antioxidant, and proapoptotic properties of saponified lipophilic *Sea buckthorn* berries extract (LSBE) in two breast cancer cell lines with different phenotypes: T47D (ER+, PR+, HER2−) and BT-549 (ER-, PR-, HER2−). The antiproliferative effects of LSBE were evaluated by an Alamar Blue assay, the extracellular antioxidant capacity was evaluated through DPPH, ABTS, and FRAP assays, the intracellular antioxidant capacity was evaluated through a DCFDA assay, and the apoptosis rate was assessed by flow cytometry. LSBE inhibited the proliferation of breast cancer cells in a concentration-dependent manner, with a mean IC_50_ of 16 µM. LSBE has proven to be a good antioxidant both at the intracellular level, due to its ability to significantly decrease the ROS levels in both cell lines (*p* = 0.0279 for T47D, and *p* = 0.0188 for BT-549), and at the extracellular level, where the ABTS and DPPH inhibition vried between 3.38–56.8%, respectively 5.68–68.65%, and 35.6 mg/L equivalent ascorbic acid/g LSBE were recorded. Based on the results from the antioxidant assays, LSBE was found to have good antioxidant activity due to its rich carotenoid content. The flow cytometry results revealed that LSBE treatment induced significant alterations in late-stage apoptotic cells represented by 80.29% of T47D cells (*p* = 0.0119), and 40.6% of BT-549 cells (*p* = 0.0137). Considering the antiproliferative, antioxidant, and proapoptotic properties of the carotenoids from LSBE on breast cancer cells, further studies should investigate whether these bioactive dietary compounds could be used as nutraceuticals in breast cancer therapy.

## 1. Introduction

Breast cancer is a heterogeneous disease caused by genetic and environmental factors and is the leading cause of cancer deaths in women worldwide [1]. Based on molecular classification, breast cancer includes the following subtypes: luminal A (ER+ and/or PR+/HER2−); luminal B (ER+ and/or PR+/HER2+); human epidermal growth factor (EGF) receptor 2 (HER2) overexpressing (ER-/HER2+); and basal-like (ER/PR−/HER−), commonly known as triple-negative (TNBC) due to its resistance to available receptor-targeted therapies. Currently, no efficient therapeutic scheme is available for TNBC management [2]. Moreover, the lack of selective cancer chemotherapeutics, especially in TNBC and chemo-resistant tumors, raises challenges in identifying new selective and/or nontoxic therapies like plant bioactive compounds that exhibit anticancer properties or can diminish the cytotoxicity of currently cytostatic therapies [3].

*Sea buckthorn* berries are one of the most nutritious and vitamin-rich fruits, containing both hydrophilic and lipophilic antioxidants [4]. *Sea buckthorn* (L.) stands among the richest sources of zeaxanthin, ranging between 19.3–42.4 mg/100 g DW, mostly in the esterified form [5], compared with 19.4 mg/100 g FW free zeaxanthin in goji berries (*Lycium barbarum* L.) [6] and 13.0 mg/100 g FW in Chinese lantern (*Physalis alkekengi* L.) [7]. β-Cryptoxanthin is important as a provitamin A xanthophyll, mostly found in esterified form in *Chinese lantern* berries (5.1 mg/100 g FW) [8], *Sea buckthorn* berries (2.1–3.8 mg/100 g DW) [5], and *Goji* berries (2.2 mg/100 g FW) [6]. β-Carotene is the most widely distributed and the most important provitamin A carotenoid. With a content of 10–20 mg/100 g DW, *Sea buckthorn* is considered a “very high (>2 mg/100 g)” source of β-Carotene [9]. The main polyphenols (phenolic acids and flavonoids) from *Hippophae rhamnoides wolongesis* range between 29.8 to 38.8 mg GAE/g, higher than that in blueberries (*Vaccinium corybosum* L.) (8.40 mg GAE/g) and blackberries (*Rubus fruticosus* L.) (7.40 mg GAE/g) [10]. The content of vitamin C found in *Sea buckthorn* berries is up to 250 mg/100g FW, followed by orange (*Citrus sinensis*) (43.61 mg/100 g) and lemon (*Citrus limon*) (31.33 mg/100 g) [10]. Among other fruits, *Sea buckthorn* has proven to possess high amounts of calcium (176.67 mg/L), followed by apricot (*Prunus armeniaca* L) (130 mg/L) and orange 110 mg/L. In addition, rich levels of iron were recorded in *Sea buckthorn* (30.9 mg/L) when compared with apricot (3.9 mg/L) and banana (*Musa* spp.) (2.6 mg/L) [11].

*Sea buckthorn* has been proven to possess many beneficial properties, such as antioxidant [4], anticancer [12], and proapoptotic [9] properties, that could help improve cancer patients’ health [13]. Moreover, several studies have demonstrated that dietary supplements rich in carotenoids improve the general health condition of cancer patients by reducing the adverse effects of anticancer therapy [14].

Considering the multitude of beneficial properties of *Sea buckthorn* and the limited number of studies investigating the bioactive properties of carotenoids from *Sea buckthorn* berries in breast cancer, the aim of the present study is to investigate, for the first time, the antiproliferative, antioxidant, and proapoptotic properties of the carotenoids from LSBE on breast cancer cells in two breast cancer cell lines with different phenotypes: T47D (ER+, PR+, HER2−) and BT-549 (ER−, PR−, HER2−).

## 2. Results

### 2.1. Chromatographic Characterization of Carotenoids from LSBE by HPLC-PDA

The total carotenoid content of the saponified extract obtained from wild-type *Sea buckthorn* berries harvested from the northwest region of Romania was 20.19 mg/100 g fresh weight (FW). Regarding the carotenoid profile, the major carotenoid pigment identified was zeaxanthin (8.61 mg/100 g), followed by *all-trans*-β-carotene (4.14 mg/100 g), lutein, *all trans*-γ-carotene, and β-cryptoxanthin, following the previous finding of our group for Romanian cultivars [15,16]. Small amounts of cis isomers of β-carotene and γ-carotene were also found, while lycopene could not be identified (Figure 1, Table 1).

Considering the potential medical application of *Sea buckthorn* extract, a saponification step was performed before cell culture tests, not only because it simplifies the chromatographic analysis of the extract, but also because in an in vivo situation xanthophyll esters present in the crude unsaponified extracts are efficiently hydrolyzed before intestinal absorption and only unesterified xanthophylls reach circulation [17].

### 2.2. Cytotoxic and Antiproliferative Effects of Total Carotenoids from LSBE and Zeaxanthin in Human Breast Cancer Cell Lines

Carotenoids have been proven to exhibit antitumor properties in a variety of cancer cell types [18]. Our results show that the TNBC cell line BT-549 proved to be the most sensitive to treatment with LSBE (IC_50_ = 12.62 µM), followed by T47D (IC_50_ = 19.40 µM) cells. In the case of zeaxanthin treatment, the IC_50_ values were close in both cell lines with an average of 75 µM (Figure 2, Table 2).

The antiproliferative effect of LSBE was statistically significant in BT-549 cells at lower concentrations starting from 15 μM, whereas zeaxanthin induced significant alterations in cell viability at higher concentrations starting from 100 μM in T47D cells (Table 3).

### 2.3. Antioxidant Activity

#### 2.3.1. Extracellular Antioxidant Capacity of LSBE

Table 4 presents the percentages of inhibition obtained after evaluating six concentrations (1000, 750, 500, 250, 100, and 50 µM) of LSBE, zeaxanthin, and ascorbic acid. The antioxidant capacity of the LSBE expressed as % ABTS inhibition varies between 3.38–56.8% and is comparable with % DPPH inhibition ranging between 5.68–68.65%. LSBE had a better inhibitory capacity than zeaxanthin in a concentration-dependent manner.

Table 5 represents the average inhibitory concentrations (IC_50_) of LSBE, zeaxanthin, and ascorbic acid needed to inhibit 50% of ABTS and DPPH. The IC_50_ values were calculated between the lowest and highest percentage of ABTS and DPPH inhibition obtained for LSBE, zeaxanthin, and ascorbic acid.

The reduction of Fe ^3+^ to Fe^2+^ by LSBE as determined by the FRAP method recorded a value of 35.6 mg/L ± 1.78 equivalent ascorbic acid/g LSBE.

#### 2.3.2. Intracellular Antioxidant Capacity of LSBE and Zeaxanthin through the DCFDA Method

A good antioxidant capacity of LSBE was observed at the intracellular level as treatment with IC_50_ concentrations of LSBE was able to significantly decrease the ROS levels in both cell lines (*p* = 0.0279 for T47D, and *p* = 0.0188 for BT-549). 

The presence of the oxidative stress conditions of LSBE treatment induced a pro-oxidant effect in T47D cells, while in BT-549 LSBE acted as an antioxidant, though the results were not statistically significant (Table 6).

According to the fluorescence intensity, in the viable cells 2′,7′-dichlorofluorescein, diacetate (DCFDA) is hydrolyzed by the intracellular esterases into a non-fluorescent compound dichlorodihydrofluorescein (DCFH), which is further transformed into a fluorescent compound 2′,7′-dichlorofluorescein (DCF) by the intracellular reactive oxygen species (ROS). The antioxidant property of LSBE is illustrated in Figure 3, where a decrease in fluorescence intensity is recorded in both cell lines. In oxidative stress conditions, however, LSBE treatment was able to reduce ROS levels only in the BT-549 triple-negative breast cancer cell line. 

### 2.4. Apoptosis Activity Evaluated through Flow Cytometry

According to flow cytometry results, treatment with IC_50_ concentrations of LSBE induced significant alterations in late-stage apoptotic cells represented by 80.29% of T47D cells (*p* = 0.0119) and 40.6% of BT-549 cells (*p* = 0.0137). These results suggest that cell death occurs mainly through apoptosis (Figure 4, Table 7).

## 3. Discussion

*Sea buckthorn* berries are known as a very good source of both hydrophilic and lipophilic bioactive compounds [19,20]. Among them, carotenoids are valuable nutrients, mainly due to their pro-vitamin A activity, antioxidant properties, and health-protective effects [21]. The carotenoid content of *Sea buckthorn* berries depends on both genetic and environmental factors, but also the degree of ripening, reaching up to 120 mg/100 g dry weight in Swedish berries [22]. In Romanian varieties, a total carotenoid content between 53–97 mg/100 g dry weight was previously reported for unsaponified extracts, while Polish cultivars contained up to 23.91 mg carotenoids/100 g fresh weight [5,19]. A recent study [23] identified a higher proportion of carotenes such as β-carotene, γ-carotene, and lycopene from Polish saponified extract, whereas our study identified zeaxanthin as the main carotenoid in LSBE.

Various in vitro studies have demonstrated that *Sea buckthorn* performs an antitumor activity in breast cancer cell lines. A previous study has shown that 0.5% ethanol/water (1:1, *v*/*v*) *Sea buckthorn* berries extract inhibits proliferation in MCF-7 breast cancer cell lines with an average of 52%. The inhibition of cancer cell proliferation was correlated with concentrations of carotenoids and vitamin C [24]. Another in vitro study discovered that procyanidins isolated from *Sea buckthorn* seeds with 70% ethanol (1:10 *w*/*v*) exert an inhibitory cell growth effect at 10–60 μg/mL concentrations (IC_50_ = 37.5 ± 1.0 µg/mL) in MDA-MB-231 cells [25]. According to the cytotoxicity assay, our results suggest that LSBE has a different antiproliferative capacity depending on the cell phenotype and morphology, indicating that the TNBC cell line BT-549 is the most sensitive to the treatment (IC_50_ = 12.62 µM), followed by luminal A phenotype T47D (IC_50_ = 19.40 µM). The IC_50_ concentrations of zeaxanthin, the main carotenoid pigment contained in our LSBE extract, ranged from 68.48 µM in BT-549 cells to 81.62 µM in T47D cells and were similar to those identified by a previous study [26] in which 62.36 µM and 92.59 µM concentrations of zeaxanthin-rich extracts obtained from saponified *Lycium barbarum* (*Goji*) were needed to inhibit the proliferation of A375 malignant melanoma cells by 50% [26]. Several methods have been used for the evaluation of the antioxidant capacity of natural compounds and plant extracts and these were classified as electron transfer-based assays (ET) or hydrogen atom transfer-based assays (HAT) [27].

For the evaluation of the antioxidant capacity of LSBE from our study, three ET-based methods have been chosen: ABTS, DPPH, and FRAP. The extracellular antioxidant activity of *Sea buckthorn* berries extract reflects its ability to capture free radicals and was also reported by several studies as follows: 1.86 mmol Trolox/100 g/dry mass for the ABTS method and 2.59 mmol Trolox/100 g/dry mass using the FRAP method for Polish *Sea buckthorn* [28]; 36.61 and 42.25 mg Trolox/g FW of scavenging activity through the DPPH method for Romanian *Sea buckthorn* [29]; 87.0–275.0 mg Trolox equivalent/g dry extract for the antioxidant activity of French *Sea buckthorn* [30]; and 60.37–79.10 mg Trolox equivalent/g extract (DPPH assay) obtained from Hungarian *Sea buckthorn* cultivars [31].

Based on the extracellular antioxidant capacity of LSBE determined through the ABTS and DPPH methods, we recorded comparable results: 3.38–56.8% of ABTS inhibition and 5.68–68.65% of DPPH inhibition. The antioxidant capacity of zeaxanthin was lower than that of LSBE with variations between 3.63–58.6% in ABTS inhibition and 3–41.75% in DPPH inhibition. Our results suggest that the antioxidant capacity varies in a concentration-dependent manner and might be influenced by tocopherols, tocotrienols, and flavonoids present in LSBE.

The variety in the antioxidant capacity of *Sea buckthorn* may be influenced by the selection of the antioxidant method (higher results were observed for DPPH than for the ABTS method) and extraction protocol. It was observed that microwave application caused the highest activity of *Sea buckthorn* berries in comparison to maceration and ultrasound [32]. However, the results of antioxidant assays vary among different studies, depending on the protocol and type of extract. Muller et al. (2011) found that *Sea buckthorn* juice had the highest lipophilic antioxidant capacity, compared with tomato juice, carrot juice, and orange juice, in all assays (DPPH, αTEAC, FRAP, LPSC), with the highest value, 738 μM TE/100 g, being obtained by LPSC (peroxyl radical scavenging assay) and the lowest by DPPH assay [33]. Phenolic compounds and flavonoids are potent scavengers of free radicals due to their hydroxyl groups and could thus represent excellent tools for antioxidant activities. The IC_50_ determined through DPPH assay for the methanolic extract from leaves of *Dittrichia viscosa* L. was 80 μg/mL; for the aqueous extract, it was 120 μg/mL. In addition, the methanolic extract had the highest capacity to scavenge ABTS+ radical (IC_50_ = 223 μg/mL), whereas the aqueous extract exhibited the lowest activity (IC_50_ = 412 μg/mL). The same study reported the highest antioxidant capacity with 944.19 mg ascorbic acid equivalent/g dw of the methanolic extract, compared with 659.441 mg ascorbic acid equivalent/g dw of the aqueous extracts as analyzed through FRAP assay [34]. Our study identified 216.12 μg/mL for IC_50_ through DPPH assay and 272.98 μg/m for IC_50_ through ABTS assay, and the results were comparable with the previous study highlighting that *Sea buckthorn* has good antioxidant potential.

The correlation between antioxidant capacity and the content of phenolic compounds was also investigated by another study where total phenolic content (16 mg GAE/g FW), isolated from Chinese *Sea buckthorn* berries, was proven to possess a high antioxidant activity of 152.5 μmol TE/g FW quantified with oxygen radical absorbance capacity (ORAC) assay, and 62.25 μmol ascorbic acid equivalent/g FW determined through peroxyl radical scavenging capacity assay (PSC) [35].

The dual roles of reactive oxygen species (ROS) in cancer cells were highlighted by a study where it was demonstrated that low ROS levels can trigger pro-tumorigenic signaling, enhancing cell proliferation and survival, while high ROS levels can boost anti-tumorigenic signaling and trigger oxidative stress-induced apoptosis of cancer cells [36]. Oxidative stress occurs when ROS production is increased and the levels of antioxidant enzymes (e.g., SOD, GPX, NADPH, GSH reductase, thioredoxin) decrease [37]. A study showed that two synthesized pyridazin-3(2h)-one derivatives, 5-(5-Propoxybenzo[b]furan-2-ylmethyl)-6-methylpyridazin-3(2H)-one and 5-[(7-Chlorobenzo[b]furan-2-yl)methyl]-6-methylpyridazin-3(2h)-thione, induced cell apoptosis via oxidative stress in a P815 murine mastocytoma cell line by deregulating the redox homeostasis due to the intracellular ROS hyper generation through significant loss of glutathione reductase and thioredoxin reductase activities [38].

The intracellular antioxidant activity of carotenoids remains controversial as there are authors that describe them both as antioxidants and pro-oxidants. However, the circumstances that define each kind of activity are very specific and particular, which makes a clear separation between the two concepts difficult [39]. The carotenoids’ pro-oxidant effects are a double-edged sword: in normal cells, they could generate oxidative damage which may decrease cell integrity and/or induce neoplastic transformation, while in tumor cells, they could induce beneficial effects like inhibition of tumor growth. At high concentrations, carotenoids were observed to act as pro-oxidants due to the formation of carotenoid ROO• and/or a faster rate of carotenoid autoxidation [40]. Indeed, in some studies, further detailed below, it is possible to observe dual activity in the same carotenoid. Using pulse radiolysis techniques, it was found that carotenoids possess a synergistic antioxidant effect along with vitamins C and tocopherol, and the pro-oxidant effect was influenced by oxygen concentration by transporting ROO• radical to lipids [41]. The enhanced levels of ROS generation have exhibited a close connection with apoptosis induced by carotenoids in a variety of cancer cell lines [42], including fucoxanthin in human leukemia HL-60 cells [43], lycopene oxidation products in MCF-7 human breast cancer cell lines [44], and lutein in HeLa cells [18]. Carotenoids are described as possessing a high antioxidant capacity due to their system of conjugated double-bond structure, which is able to delocalize unpaired electrons [45].

To our knowledge, no prior literature has reported data about the general ROS antioxidant capacity of LSBE in breast cancer cells. Nevertheless, a study reported that treatment with 10 µM β-cryptoxanthin slightly reduced ROS levels in HeLa and MDCK cells after oxidative stress induction with H_2_O_2_, and it was observed that under physiological conditions with higher levels of basal ROS and high oxygen tension, the scavenging capacity of β-cryptoxanthin decreased and acted as a pro-oxidant molecule, thus resulting in further elevation of ROS levels and triggering the oxidative stress-induced apoptosis of HeLa cells [18]. This dynamic control of ROS production is most likely modulated by both the pro-oxidant capacity and the antioxidant capacity of the carotenoids [46]. At low oxygen pressures, carotenoid molecules can act as strong oxidants, while at high O2 pressures, the carotenoids oxidize rapidly thus exerting pro-oxidant activities [47]. Additionally, it was found that, in the absence of oxidative stress conditions, treatment with 50 μM lycopene induced 4.3% ROS production in the MCF-7 breast cancer cell line when evaluated by flow cytometry, using 2′,7′-dichlorofluorescein diacetate (DCFH-DA) as a probe [19]. The pro-oxidant effects of carotenoids have also been reported in vitro when high concentrations were used. This effect may be explained by a more favorable formation of carotenoid ROO• and/or by a faster rate of carotenoid autoxidation [40]. It was discovered that astaxanthin works synergistically with β-carotene and lutein to trigger ROS production and apoptosis in MCF-7 cells, whereas the IC_50_ and combination-index values of astaxanthin co-treatment with a lower concentration of β-carotene and lutein (5 μM) exhibited enhanced cytotoxicity and oxidative stress as compared with individual carotenoids or saponified carotenoid extract from shrimp [48]. Moreover, another study revealed that treatment with 2 µM lutein increased ROS levels by 1.9-fold in MCF-7 and MDA-MB-468 breast cancer cells compared to control, and with the addition of radical oxygen scavenger N-acetyl cysteine, the growth inhibition effect induced by lutein was attenuated, suggesting that ROS production induced by lutein plays an important role in the growth inhibitory effect of breast cancer cells [49].

The results from the present study indicate that the carotenoids from LSBE acted as antioxidants at the IC_50_ concentrations specific to each breast cancer cell line (20 µM in T47D and 13 µM in BT-549) by significantly decreasing the ROS levels (*p* = 0.0279 in T47D and *p* = 0.0188 in BT-549), while zeaxanthin was able to slightly reduce the ROS levels in both cell lines without significant statistical results. Moreover, ROS levels in oxidative stress conditions (H_2_O_2_-treated cells) were slightly reduced by the LSBE treatment only in BT-549 triple-negative breast cancer cell lines, whereas LSBE acted as a pro-oxidant in T47D cells. Our results suggest that the anti/pro-oxidant activity of total carotenoids isolated from LSBE depends on the phenotype of each cell line, the concentration of carotenoids, and the treatment time.

It is currently believed that the inhibition of proliferation represents not only the measure of antitumor treatment efficacy but also the ability to induce apoptosis in tumor cells. Apoptosis is a genetically programmed process of cell destruction, without loss of integrity, causing lysis or inflammation without damaging adjacent tissue cells [50]. To our knowledge, no prior literature has reported data on apoptotic cell death induced by treatment with LSBE in breast cancer cell lines. Nevertheless, a previous study mentioned above found that carotenoid lutein induced minimal apoptotic cell death in breast cancer cells MDA-MB-468 and MCF-7 treated with 2 µM lutein for 24 h, having a 5.21% of early-stage apoptotic (annexin V+/PI−) population in MDA-MB-468 cells and a minor increased (<10%) late-stage apoptotic/necrotic (annexin V+/PI+) cell fraction in MDA-MB-468 cells, but not in MCF-7 cells [49]. Our flow cytometry results show that cell death was induced by treatment with IC_50_ concentrations of LSBE with significant alterations in late-stage apoptotic cell population represented by 80.29% of T47D cells (*p* = 0.0119) and 40.6% of BT-549 cells (*p* = 0.0137). Likewise, treatment with IC_50_ concentrations of zeaxanthin in T47D cells indicates that apoptosis occurs mainly throughout late-stage (10.68%), while in BT-549 cells a slightly increased percentage (5.12%) of the early-stage apoptotic population was reported, compared to late-stage apoptotic cells (4.14%). Overall, the LSBE induced a stronger apoptosis effect than zeaxanthin in both breast cancer cell lines.

Overall, the present study demonstrates that lipophilic *Sea buckthorn* extract (LSBE) possesses both anticancer and antioxidant properties; however, it presents some limitations. Many in vitro studies evaluate the anticancer properties of plant-based bioactive compounds using one [51,52] or more different cell lines [53,54] for a certain cancer pathology and at least three biological replicates (n = 3). The relevance of our study was also confirmed by the selected methods in order to evaluate the anticancer properties at the cellular level as follows: the Alamar blue assay [55] used to test the cytotoxic and antiproliferative effect of a compound is not cytotoxic and is more sensitive than tetrazolium assays (e.g., MTT, XTT, WST-8), while the apoptosis assay through flow cytometry [56] is more accurate when determining the precise number of apoptotic cell populations than immunocytochemistry where cells may suffer significant damage from the paraformaldehyde used in the fixation protocol. Although our study selected two different cell lines with different phenotypes and performed the experiments in three biological replicates (n = 3), the study should be extended to more breast cancer cell lines with different phenotypes as well to healthy cells, in order to prove the cytotoxic selectivity and proapoptotic properties of LSBE. Regarding the evaluation of the intracellular antioxidant activity of LSBE, the selected method for our study was based on 2′,7′-dichlorodihydrofluorescein diacetate (DCFH-DA), the most widely used method for detecting total reactive oxygen species (ROS) [57] and for testing the antioxidant capacity of a plant-based extract both in cancer and healthy cells [58]. Concerning the extracellular antioxidant activity of LSBE, our study selected three of the most efficient and frequently used methods based on electron transfer (DPPH, ABTS, and FRAP), which are representative of determining the antioxidant capacity of an extract to reduce free radicals. Although many studies have used free radical scavenging assays like DPPH [59], ABTS [60], and FRAP [61] in order to evaluate the antioxidant property of a plant-based extract, our study extends to investigating the activities of relevant antioxidant enzymes (e.g., catalase, peroxidase, superoxide dismutase), as well as measuring specific markers like H_2_O_2_ and malondialdehyde levels in order to confirm the antioxidant power of LSBE.

Further in vitro studies are needed to elucidate the mechanism of action of each carotenoid in the apoptosis extrinsic or intrinsic pathway, whether activated through high ROS levels or by other mechanisms, as well as their role as pro- or antioxidants. Molecular studies that investigate specific genes that regulate apoptosis, inhibit tumor growth, or activate the pro-or antioxidant capacity are also needed. However, to prove the efficiency of daily dietary *Sea buckthorn* extracts to prevent tumor development and boost the immune system, complex in vivo studies are needed.

## 4. Materials and Methods

### 4.1. Sample Collection and Preparation

*Sea buckthorn* (*Hippophae rhamnoides* L.) wild berries were harvested from Aghiresu commune, Cluj County, in the northwest region of Romania in August 2020. All the berries were frozen and stored at −20 °C in LDPE (low-density polyethylene) bags until further use. Freshly thawed berries were manually separated from the seeds and homogenized using an Ultraturax homogenizer.

### 4.2. Isolation of Total Carotenoids from Sea buckthorn Berries

The carotenoids were exhaustively extracted from *Sea buckthorn* berries with a mixture of petroleum ether: methanol and ethyl acetate (1:1:1, *v*/*v*/*v*). The combined extracts were filtered and then partitioned in a separation funnel with diethyl ether and saturated NaCl solution. The upper organic phase was collected, dried over anhydrous sodium sulfate, and evaporated until dry. Samples were stored at −20 °C until they were subjected to saponification. The samples were dissolved in an appropriate volume of diethyl ether for saponification. An equal volume of 30% potassium hydroxide solution (in methanol) was added and the sample was stirred for 6 h, in the dark, at room temperature, under nitrogen. The mixture was transferred into a separation funnel containing diethyl ether, washed with 5% NaCl until alkali-free, and concentrated until dry. The *Sea buckthorn* extracts were dissolved in ethyl acetate and filtered through 0.2 μm PTFE filters before HPLC analysis. All the experiments, extraction, saponification, and HPLC analysis were performed three times, using the same batch of berries.

### 4.3. HPLC-DAD Analysis of Total Carotenoids

HPLC-DAD separation was performed using a Shimadzu LC20 AT HPLC system (Shimadzu Corporation, Kyoto, Japan) with an SPDM20A diode array detector and a YMC C30 reversed-phase column (250 mm length, 4.6 mm inner diameter and 5 μm particle size). The experimental conditions for the separation and identification by HPLC-DAD were the same as described in a previous study [62]. Quantification of carotenoids was performed using external calibration with standards of β-carotene, lutein, β-cryptoxanthin, and zeaxanthin purchased from Extrasynthese (Lyon, France) in the range of 1–100 μg/mL. The HPLC-DAD analysis was performed three times for the tested sample and data are expressed in mg/100 g F.W (fresh weight) and presented as the mean ± SD of these three measurements.

### 4.4. Cell Lines and Culture Conditions

The human breast cancer cell lines T47D (ductal carcinoma, epithelial subtype, ER+, PR+, HER− from ECACC) and BT-549 (ductal carcinoma, mesenchymal subtype, ER−, PR−, HER2−, from ATCC) were used in the experimental design of this study. The cell lines were cultured in RPMI-1640 medium supplemented with 10% FBS, 1% penicillin-streptomycin, 1% glutamine, and 0.023% insulin for BT-549 (0.2% insulin for T47D). All the cell lines were incubated at 37 °C in a humidified atmosphere of 95% air and 5% CO_2_. All cell culture reagents were purchased from Sigma-Aldrich (St. Louis, MO, USA).

### 4.5. Cytotoxicity Assay

Cell viability was determined after 24 h by Alamar Blue Assay based on the conversion of resazurin (non-fluorescent blue dye) to resorufin (pink fluorescent) by mitochondrial enzymes in viable cells. The comparison between the Alamar blue absorbance measured in the presence or absence of treatment with LSBE and zeaxanthin reflects antiproliferative capacity. The cells were seeded into 96-well plates at a density of 2 × 104 cells/mL in 200 µL medium and left 24 h to attach. After 24 h of incubation, the culture medium was removed and the cells were treated with 100 µL of fresh culture medium containing 10% FBS, six different concentrations of LSBE (1, 10, 15, 25, 50, and 100 μM), or six different concentrations of zeaxanthin (1, 10, 50, 100, 150, and 200 μM), whereas control cells were treated with the specific culture medium containing 0.5% DMSO and 10% FBS. Doxorubicin, the most commonly used drug in breast cancer chemotherapy, was used as the positive control (1–16 μM) [63]. After 24 h of incubation with treatment, 10 µL of Alamar Blue (Thermo Fisher Scientific, Waltham, MA, USA) reagent was added to each well, and absorbance was measured at 570 nm and 600 nm wavelength using a multi-mode reader (Synergy HTX, BioTek, Charlotte, VT, USA) [64].

For consistency, we have chosen to express the concentration of all the tested substances (Zeaxanthin, Doxorubicin, H_2_O_2_), but also that of the LSBE, in terms of molarity. Although biological samples (plasma, tissue, etc.) contain mixtures of carotenoids, their total concentration is commonly expressed in terms of molarity, from carotenoid intake to carotenoid blood and tissue concentrations—implications for dietary intake recommendations [65]. For the calculation of the molar concentration of total carotenoids from LSBE, we used a weighted average molecular mass, according to the results obtained by HPLC analysis. As LSBE is a mixture of carotenoids, we determined by HPLC the amount of each carotenoid in the sample (external calibration). We then calculated the mass percentage % of each carotenoid in the mixture (Table 1). Furthermore, we calculated the weighted average molecular mass/weight of total carotenoids, taking into account their percentage in the mixture and their specific molecular weight. We obtained a value of 557.75 g/mol for the carotenoid mixture of LSBE. Using this value, a 1 µM LSBE concentration in the culture media corresponds to 0.557 µg LSBE/mL.

Cell viability was expressed as a percentage of the control and calculated according to the following formula: cell viability (%) = [(O.D. treated cells at 570 nm − O.D. treated cells at 600 nm) − (O.D. medium without cells at 570 nm − O.D. medium without cells at 600 nm)]/[(O.D. untreated cells at 570 nm − O.D. untreated cells at 600 nm) − (O.D. cell-free medium at 570 nm − O.D. cell-free medium at 600 nm) × 100. The half maximal inhibitory concentration (IC_50_) values of the two human breast cancer cell lines were calculated using the log (inhibitor) vs. normalized response-variable slope in GraphPadPrism Software Version 8 (GraphPad Software, Inc. Avenida de la Playa La Jolla, San Diego, CA, USA). The cytotoxicity assay was performed for three different cell passages considered biological replicates. We used 4 technical replicates for each treatment in each cell passage and a mean was calculated for each of the 4 technical replicates. Final data are presented as the mean ± SD of the biological replicates (n = 3).

### 4.6. Antioxidant Capacity

To evaluate the antioxidant activity of LSBE we selected the following three assays: ABTS (Tocopherol equivalent antioxidant capacity), which uses the radical cation ABTS+; DPPH, based on the free radical 2,2-diphenyl-1-picrylhydrazyl scavenging capacity; and FRAP (Ferric reducing antioxidant power). Thus, to determine the antioxidant capacity of LSBE, we used an established method [66] and modified it as follows. To obtain the antioxidant extract in the hydrophilic form, 5 mL of methanol/water solution (80:20, *v*/*v*) was added to 5 g of LSBE. The mixture was vortexed for 1 min and then centrifuged at 5000 rpm for 7 min. The alcoholic supernatant with antioxidant components was used for the antioxidant tests.

#### 4.6.1. ABTS Assay

This antioxidant test is based on the ability of the antioxidants to reduce the activity of ABTS + cation, a blue-green chromophore that absorbs at 734 nm. ABTS + is produced by the reaction between ABTS (2,2-azinobis-(3-ethylbenzothiazoline)-6-sulfonic acid) stock solution with potassium persulphate (K2S2O8). To perform this antioxidant test, we used a method previously described [67] with the following modifications: 50 µL of LSBE alcoholic extract was added to 2.450 mL of 7 mM ethanolic solution of ABTS and the mixture was vortexed in the dark, at an ambient room temperature for 6 min. Afterward, the absorbance was measured at 734 nm by using a UV-VIS spectrophotometer, model V 530 (JASCO, Oklahoma City, OK, USA). The results were expressed as % inhibition using the following formula: % inhibition = [(T0 − T6)/T0] × 100, where T0 represents the absorption at time zero and T6 represents the absorption at 6 min. The half maximum inhibitory concentration (IC50) was calculated using GraphPadPrism Software Version 8 (GraphPad Software, Inc. Avenida de la Playa La Jolla, San Diego, CA, USA), by linear regression analysis curve plotting between inhibition percentage and concentration of LSBE and zeaxanthin. Ascorbic acid was used as the positive control. The ABTS assay was performed for six concentrations. Three technical replicates were used for each concentration. Data are presented as % ABTS inhibition of the three measurements ± SD (n = 3).

#### 4.6.2. DPPH Assay

The DPPH antioxidant assay was performed according to an established method [68], modified as follows: LSBE was diluted with ethyl acetate (1:10, *v*/*v*), then 2 mL of a 10^−4^ M DPPH● stock solution previously prepared with ethyl acetate was added to 500 μL of diluted LSBE. An absorbance at 515 nm was read immediately after the addition of the (T0) and after 30 min of incubation (T30). The measurements were taken using a UV-VIS spectrophotometer, model V 530 (JASCO, Oklahoma City, OK, USA), and the results were expressed as inhibition percentage and calculated using the following formula: % inhibition = [(T0 − T30)/T0] × 100, where T0 represents the absorption at time zero and T30 represents the absorption at 30 min. The half maximum inhibitory concentration (IC50) was calculated using GraphPadPrism Software Version 8 (GraphPad Software, Inc. Avenida de la Playa La Jolla, San Diego, CA, USA), by linear regression analysis curve plotting between inhibition percentage and concentration of LSBE, zeaxanthin, and ascorbic acid. Ascorbic acid was used as the positive control. The DPPH assay was performed for six concentrations. Three technical replicates were used for each concentration. Data are presented as % DPPH inhibition of the three measurements ± SD (n = 3).

#### 4.6.3. FRAP Assay

Another simple and automated test that measures the antioxidant power of an extract or compound is the capacity of ferric reduction (FRAP) method. The FRAP method consists of the reduction of ferric ions to ferrous ions at low pH resulting in the formation of a colored ferrous complex called tripyridyl triazine. FRAP values are obtained by comparing the modifications of the tested reaction mixtures with those containing ferrous ions of known concentration at an absorbance of 595 nm. The absorption modifications are linear over a wide range of concentrations containing mixtures of antioxidants, including those from blood plasma from solutions containing antioxidants in purified form. There is no apparent interaction between the antioxidants. The FRAP antioxidant assay was performed according to an established method [69]. The following working solutions were prepared: 300 mM acetate buffer with pH = 3.6, 10 mM TPTZ (2,4,6-Tripyridyl-s-triazine) in 40 mM HCl and 20 mM FeCl_3_ × 6 H_2_O. The FRAP solution was freshly prepared by mixing 10 mL of acetate buffer with 1 mL of TPTZ solution and 1 mL of FeCl_3_ × 6 H_2_O solution. A total of 100 µL of diluted LSBE, like in the DPPH method described above, was added to 500 µL FRAP solution and 2 mL of double distilled water and incubated in the dark at room temperature for one hour. Afterward, the absorbance was measured as 595 nm using a UV-VIS spectrophotometer, model V 530 (JASCO, Oklahoma City, OK, USA). Ascorbic acid was used as the positive control. The FRAP assay was performed three times for the tested sample. Data are expressed in mg/L equivalent ascorbic acid/g LSBE and presented as the mean ± SD of these three measurements.

#### 4.6.4. General ROS Assay

The quantification of intracellular reactive oxygen species (ROS) uses the fluorescent probe 2′,7′-dichlorodihydrofluorescein diacetate, which is cell membrane permeable. It is hydrolyzed by cellular esterases to 2′,7′-dichlorofluorescein, which further reacts with intracellular ROS to form fluorescent 2′,7′-dichlorofluorescein. The cell lines were grown in the culture medium specific to each cell line. When the confluence in the culture flask reached 70–90%, the cells were detached with a 0.25% trypsin-EDTA solution and seeded in 96-well microplates. Several 2 × 104 cells/well using 5 wells for each treatment were incubated overnight at 37 °C and 5% CO_2_. After 24 h of incubation, the oxidative stress induction treatment was performed for 15 min with 500 µM H_2_O_2_ (Honeywell, Fluka, Wabash, IN, USA). After the induction of oxidative stress, the cell culture medium was replaced and the cells were treated for 1 h with 1 μM physiological concentrations of LSBE and zeaxanthin, with concentrations close to IC_50_ values of LSBE and zeaxanthin specific to each cell line. Afterward, the culture medium was removed and the cells were washed 1–2 times with PBS with CaCl_2_ and MgCl_2_ (Sigma Aldrich, St. Louis, MO, USA) and incubated for 15 min with 10 µM 2′,7′-dichlorodihydrofluorescein diacetate (DCFDA). The fluorescence was measured at 1 h using a Biotek Synergy HTX (Santa Clara, CA, USA) multimode microplate reader at 485/20 nm excitation and 528/20 nm emission. H_2_O_2_ was used as the positive control in the general ROS assay. The general ROS assay was performed for three different biological replicates for untreated and treated cells. We used 4 technical replicates for each treatment in each cell passage and a mean was given for each of the 4 technical replicates. Final data are expressed as % ROS inhibition relative to control (untreated cells, respectively cells treated with H_2_O_2_) and presented as the mean ± SD of biological replicates (n = 3).

### 4.7. Apoptosis Rate Evaluation through Flow Cytometry

A total of 5 × 10^5^ cells/well were seeded in 6-well culture plates with 3 mL of a specific culture medium for each cell line. After 24 h of incubation at 37 °C, 5% CO_2_ the cells were treated with physiological concentrations of 1 μM of LSBE and zeaxanthin, with concentrations close to IC_50_ values of LSBE and zeaxanthin specific to each cell line as described above. After 4 h of treatment, the cells were harvested, labeled with Annexin V-FITC and propidium iodide (PI), and sorted using the S3e Cell Sorter (Bio-Rad, Becton Dickinson, Franklin Lakes, NJ, USA) flow cytometer according to the protocol described below. The Annexin V-FITC and propidium iodide apoptosis kit (Thermo Fisher Scientific, Waltham, MA, USA) was used. The cells were detached from the 6-well plate by pipetting 300 μL of trypsin-0.25% EDTA (Sigma-Aldrich) and inactivated with 3 mL of fresh medium. The cells were centrifuged at 2000 rpm for 5 min and resuspended in 1 mL of cold PBS (Sigma Aldrich, St. Louis, MO, USA). After centrifugation for 5 min at 1000 rpm, the cells were suspended in 100 μL binding buffer 1× mixed with 5 μL of Annexin V-FITC dye and 1 μL of propidium iodide (100 μg/mL). After 15 min of incubation at room temperature, 400 μL of 1× binding buffer was added. The samples were kept on ice until they were evaluated using an S3e Cell Sorter flow cytometer (Bio-Rad) with FACS Diva program version 6.1.3 (Becton Dickinson, Franklin Lakes, NJ, USA). Apoptotic data were reported as the percentage of apoptosis, obtained by determining the number of apoptotic cells versus the total number of cells. For the apoptosis assay, three biological replicates for each group were used. Data are expressed as % of early- and late-stage apoptotic cells, or necrotic cells, and presented as the mean ± SD of the biological replicates in each group (n = 3).

### 4.8. Image Acquisition and Processing of 2′,7′-Dichlorofluorescein Diacetate (DCFDA) Stained Cells

Sample visualization in fluorescence microscopy of intracellular reactive oxygen species (ROS) activity in cells exposed to 1 μM physiological concentrations of LSBE and zeaxanthin, with the IC_50_ concentrations of LSBE and zeaxanthin specific to each cell line, was performed using a Zeiss Axiovert D1 inverted phase microscope, equipped with a 10× objective and 488 nm filter. The fluorescence images were taken with an AxioCam MRc camera, and the image processing and analysis were done with Axiovision Rel 4.6 morphometry software (Carl Zeiss, Jena, Germany).

### 4.9. Statistical Analysis

The Shapiro-Wilk test was used to assess the normality of the data distribution. The differences in cell viability (OD), ROS, and apoptosis data between controls and different concentrations of LSBE and zeaxanthin were tested with the Kruskal-Wallis test, followed by Dunn’s multiple comparison post hoc test, according to data distribution. The statistical analysis was performed in GraphPad Prism Software Version 8 (GraphPad Software, Inc., Avenida de la Playa La Jolla, San Diego, CA, USA). A *p*-value less than 0.05 was considered significant.

## 5. Conclusions

In the present work, we demonstrate for the first time that the lipophilic *Sea buckthorn* extract (LSBE) exhibits both antioxidant and anticancer properties. Our study indicates that, under experimental conditions, LSBE exerts an anti-cancer effect by cell proliferation inhibition and apoptosis induction. Moreover, LSBE has proven to possess a good antioxidant potential both intracellularly and extracellularly in a dose-dependent manner. Considering that the extraction method used in this study is highly relevant to obtain enriched carotenoid-based fraction, we cannot exclude the presence of other bioactive compounds that can potentate the investigated bioactive properties. Overall, the data from this study provide experimental evidence supporting the potential use of *Sea buckthorn* berries as a functional bioactive ingredient in breast cancer complementary therapy. Based on these findings, further studies on the molecular pathways underlying the demonstrated bioactive properties are required in order to confirm that carotenoid intake can reduce breast cancer risk.

## Figures and Tables

**Figure 1 molecules-28-04486-f001:**
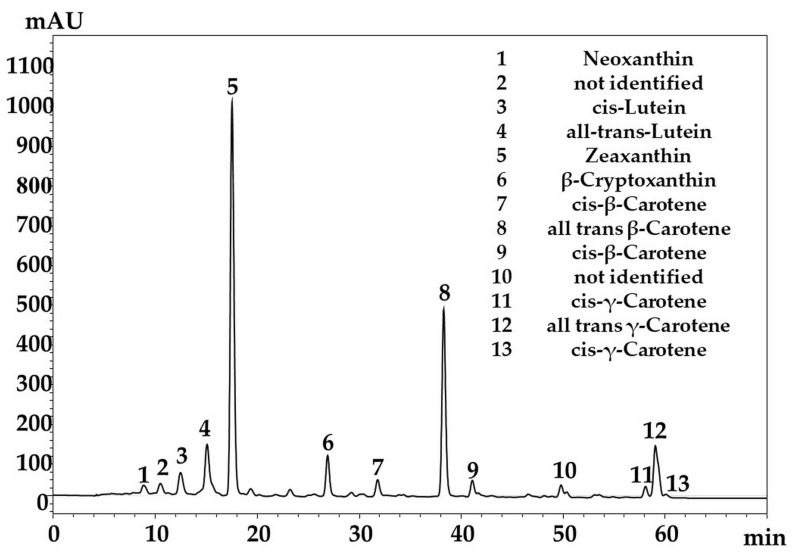
The C30-HPLC-PDA (450 nm) chromatogram of carotenoids from saponified *Sea buckthorn* extract.

**Figure 2 molecules-28-04486-f002:**
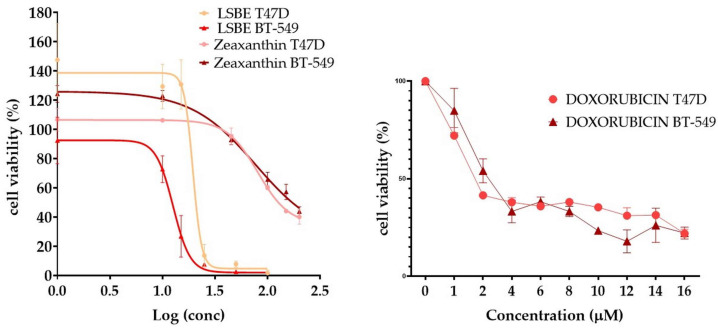
The cytotoxic activity of LSBE, zeaxanthin (ZEA), and doxorubicin (DOXO) on T47D and BT-549 breast cancer cell lines at 24 h of treatment (mean ± SD, n = 3).

**Figure 3 molecules-28-04486-f003:**
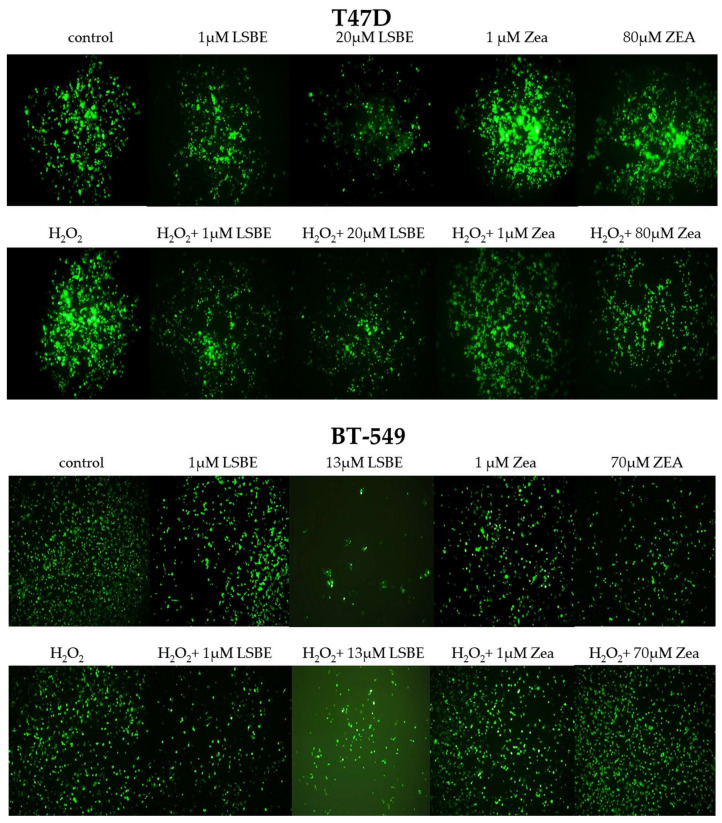
The effect of LSBE and zeaxanthin on the production and accumulation of reactive oxygen species (ROS) in T47D and BT-549 breast cancer cell lines at 1 h. Cells were observed in fluorescence using a 488 filter under an inverted fluorescence microscope at 10× magnification.

**Figure 4 molecules-28-04486-f004:**
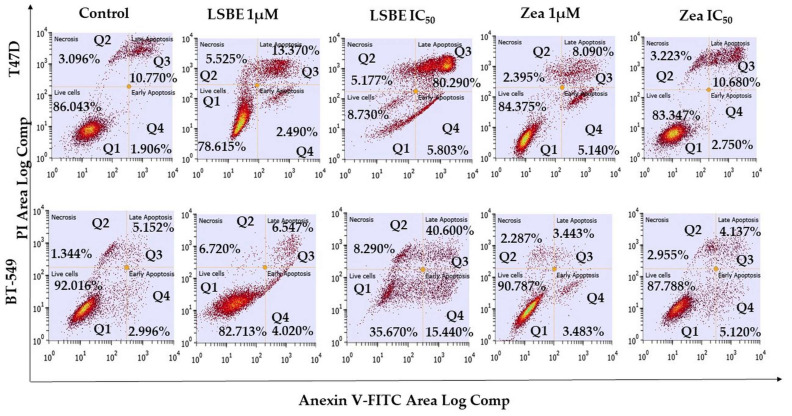
Detection of apoptosis in T47D and BT-549 cells treated with LSBE and zeaxanthin and analyzed by flow cytometry using annexin V-FITC and propidium iodide (PI) staining. Viable (non-apoptotic) cells are annexin V-FITC- and PI-negative (Q1), annexin V-FITC-positive and PI-negative cells are in early apoptosis (Q4), both annexin V-FITC- and PI-positive cells are in late apoptosis (Q3), and necrotic cells are annexin V-FITC-negative and PI-positive (Q2).

**Table 1 molecules-28-04486-t001:** The spectral characteristics and the carotenoid content (mg/100 FW) of saponified *Sea buckthorn* berries extract. The results are presented as the mean of three measurements ± SD of the same sample (mean ± SD, n = 3).

ID	Identification	UV-Vis Maxima	Concentration (mg/100 g F.W)	% of Total Carotenoids
1	Neoxanthin	416,439.468	0.40 ± 0.07	1.96
2	not identified	400, 422, 448	0.39 ± 0.04	1.92
3	*cis*-Lutein	330, 420, 441, 472	0.80 ± 0.16	3.96
4	*all-trans-*Lutein	422, 444, 473	1.80 ± 0.43	8.93
5	Zeaxanthin	427, 450, 477	8.61 ± 0.81	42.62
6	β-Cryptoxanthin	428, 451, 476	0.94 ± 0.22	4.64
7	*cis*-β-Carotene	338, 420, 449,472	0.49 ± 0.19	2.42
8	*all trans* β-Carotene	421, 451, 478	4.14 ± 0.23	20.52
9	*cis*-β-Carotene	345, 421, 447, 473	0.39 ± 0.13	1.95
10	not identified	420, 441, 465	0.28 ± 0.11	1.42
11	*cis*-γ-Carotene	361, 433, 460, 491	0.26 ± 0.09	1.31
12	*all trans* γ-Carotene	434, 461, 492	1.65 ± 0.21	8.15
13	*cis*-γ-Carotene	358, 431, 458, 489	0.04 ± 0.03	0.18
	Total		20.19 ± 2.72	

**Table 2 molecules-28-04486-t002:** The cytotoxicity of LSBE, zeaxanthin, and doxorubicin was expressed as half inhibitory concentration (IC_50_, sigmoidal dose-response), and the antiproliferative capacity was expressed as significant negative hillslope derived from the linear regression of time-dependent inhibition (mean ± SD, n = 3).

Treatment	LSBE		Zeaxanthin		Doxorubicin	
Cell Line	T47D	BT-549	T47D	BT-549	T47D	BT-549
IC_50_ (μM)	19.45	12.62	81.62	68.48	1.774	3.183
IC_50_ (µg/mL)	34.87	22.63	143.47	120.37	3.06	5.49
HillSlope	−10.44	−5.357	−3.403	−1.909	−0.3771	−0.743
logIC_50_	1.289	1.101	1.912	1.836	0.5028	0.248
R^2^	0.9132	0.9014	0.9514	0.9798	0.900	0.8904
*p*-value	<0.0001	<0.0001	<0.0001	<0.0001	<0.0001	<0.0001

**Table 3 molecules-28-04486-t003:** The antiproliferative effect of LSBE and zeaxanthin on cell viability at 24 h of treatment in T47D and BT-549 breast cancer cell lines (n = 3).

Treatment				LSBE		Zeaxanthin (ZEA)		Doxorubicin (DOXO)	
Cell Line				T47D	BT-549	T47D	BT-549	T47D	BT-549
Concentration				*p*-Value	*p*-Value	*p*-Value	*p*-Value	*p*-Value	*p*-Value
(μM)	LSBE (µg/mL)	ZEA (µg/mL)	DOXO (µg/mL)						
1	1.79	1.76	1.72	>0.9999	>0.9999	>0.9999	0.6252	>0.9999	>0.9999
2			3.45					>0.9999	>0.9999
4			6.90					0.3178	0.1003
6			10.35					0.0970	0.4696
8			13.79					0.2557	0.1932
10	17.93	17.58	17.24	>0.9999	>0.9999	>0.9999	0.7982	0.0266	0.0007
12			20.69					0.0004	0.0005
14			24.14					0.0001	0.0462
15	26.89			>0.9999	0.0407				
16			27.59					0.0012	0.0149
25	44.82			0.1745	0.0030				
50	89.65	87.89		<0.0001	<0.0001	>0.9999	>0.9999		
100	179.29	175.78		<0.0001	<0.0001	0.0292	0.3206		
150		263.67				0.0001	0.0459		
200		351.56				<0.0001	0.0015		

**Table 4 molecules-28-04486-t004:** The percentage of DPPH and ABTS inhibition induced by ascorbic acid as compared to LSBE and zeaxanthin (mean ± SD, n = 3).

Concentration				% ABTS Inhibition						% DPPH Inhibition					
(µM)	(µg/mL)			Ascorbic Acid (AA)		LSBE		Zeaxanthin (ZEA)		Ascorbic Acid		LSBE		Zeaxanthin	
	AA	LSBE	ZEA	Mean	SD	Mean	SD	Mean	SD	Mean	SD	Mean	SD	Mean	SD
1000	176	557.75	568.9	92.8	0.57	56.80	0.28	58.60	0.85	87.13	1.24	68.65	0.07	41.75	1.06
750	132	418.31	426.7	65.7	0.99	46.88	0.45	42.55	0.78	60.73	0.39	52.90	0.28	32.13	1.59
500	88	278.874	284.49	43.3	0.42	37.21	0.30	30.60	0.85	42.50	0.71	37.05	0.07	23.75	0.35
250	44	139.42	142.225	27.2	0.28	13.80	0.64	14.65	0.49	23.85	0.21	23.57	0.62	14.25	0.49
100	17.6	55.775	56.89	12.2	0.28	7.48	0.25	7.58	0.11	10.90	0.57	10.87	0.47	5.55	0.35
50	8.8	27.88	28.45	5.3	0.14	3.38	0.11	3.63	0.11	6.13	0.18	5.68	0.11	3.00	0.28

**Table 5 molecules-28-04486-t005:** The concentrations of the inhibitors (ascorbic acid, LSBE, zeaxanthin) at which the response is reduced by half (IC_50_).

Compound	IC_50_ ABTS			IC_50_ DPPH		
	% Inhibition	µM	µg/mL	% Inhibition	µM	µg/mL
Ascorbic acid	46.43	500.42	88.05	43.8	502.7	88.48
LSBE	25.92	456.45	272.98	32.14	468.23	261.12
Zeaxanthin	27.8	490.18	262.16	20.05	480.23	279.34

**Table 6 molecules-28-04486-t006:** Intracellular reactive oxygen species level in human breast cancer T47D and BT-549 cells treated with LSBE and zeaxanthin, as determined by the fluorescence test with 2′,7′-dichlorofluorescein (DCF) (mean ± SD, n = 3).

DCF Fluorescence (% of Control)						
T47D				BT-549		
Treatment	Mean	SD	*p*-Value	Mean	SD	*p*-Value
control	100	0	-	100	0	-
LSBE 1 µM	89.36	12.15	0.1194	81.89	15.93	0.0725
LSBE IC_50_	79.22	21.36	0.0279	43.86	48.05	0.0188
Zeaxanthin 1 µM	94.58	4.075	0.1694	94.60	5.875	0.3815
Zeaxanthin IC_50_	99.21	3.489	0.7140	91.56	13.29	0.2311
H_2_O_2_	100	0	-	100	0	-
H_2_O_2_+ LSBE 1 µM	104.6	11.08	0.7834	94.36	8.747	0.6140
H_2_O_2_+ LSBE IC_50_	101.7	23.37	0.7834	70.23	31.57	0.3592
H_2_O_2_+ Zeaxanthin 1 µM	103.7	20.43	0.8546	99.63	3.932	0.6466
H_2_O_2_+ Zeaxanthin IC_50_	103.5	12.35	0.8546	121.8	35.45	0.3356

**Table 7 molecules-28-04486-t007:** The effect of LSBE and zeaxanthin on apoptosis in T47D and BT-549 breast cancer cell lines evaluated by flow cytometry (mean ± SD, n = 3).

Treatment	Early Apoptosis (%)			Late Apoptosis (%)			Necrosis (%)		
**T47D**									
	**Mean**	**SD**	***p*-Value**	**Mean**	**SD**	***p*-Value**	**Mean**	**SD**	***p*-Value**
control	2.620	3.200	-	7.367	5.398	-	3.970	3.294	-
LSBE 1 µM	2.490	3.479	0.9626	13.370	7.821	0.3988	5.525	3.500	0.4255
LSBE 20 µM	5.803	3.278	0.2084	80.290	8.922	0.0119	5.177	2.988	0.5294
Zeaxanthin 1 µM	5.140	7.212	0.4255	8.090	6.746	0.6731	2.395	1.563	0.5422
Zeaxanthin 80 µM	2.750	4.254	>0.9999	10.680	6.891	0.5294	3.223	1.977	0.7532
**BT-549**									
	**Mean**	**SD**	***p*-Value**	**Mean**	**SD**	***p*-Value**	**Mean**	**SD**	***p*-Value**
control	3.370	2.675	-	2.857	0.2272	-	1.757	0.8977	-
LSBE 1 µM	4.020	1.438	0.8551	6.547	0.9659	0.2012	6.720	8.011	0.3798
LSBE 13 µM	15.440	9.525	0.0552	40.600	31.19	0.0137	8.290	8.048	0.3291
Zeaxanthin 1 µM	3.483	1.926	>0.9999	3.443	2.309	0.9273	2.287	1.181	0.6256
Zeaxanthin 70 µM	5.120	1.779	0.2733	4.137	3.252	0.6481	2.955	3.656	0.8273

## Data Availability

The data are contained within this article.

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
