# Peer review of "The Bioactive Properties of Carotenoids from Lipophilic Sea buckthorn Extract (Hippophae rhamnoides L.) in Breast Cancer Cell Lines"

_molecules, 2023, doi:10.3390/molecules28114486_

Round 1

Reviewer 1 Report

The paper titled 'The Bioactive Properties of Carotenoids from Lipophilic Sea Buckthorn Extract (Hippophae rhamnoides L.) in Breast Cancer Cell Lines' is an good contribution to the field of cancer research. The authors have investigated the bioactive properties of carotenoids extracted from sea buckthorn and their potential anti-cancer effects on breast cancer cell lines. However, there are some points that need to be considered to improve the quality of the manuscript

Line 21-22: the sentence is not clear, please reformulate

Line 24: please add cells after T47D

Line 24-26: the sentence is not clear, please reformulate

Line 45: please add a reference. You can consider https://doi.org/10.3390/app12041866

Line 92: please add ‘of’ to the sentence : proapoptotic activity of…

Line 112: in-vivo in italic

Line 119: you expressed the concentration of LSBE in molarity. How did you calculate this concentration ? The same for table 2 and other statements.

Line 133-136: Could you please rephrase the sentence and improve the language to make it more understandable?

The results for the Extracellular antioxidant capacity are missing. Please include the results and express them for the ABTS and DPPH activities as IC50 values

Line 177-279: please rewrite

Line 204-209: this is a long sentence that should be rewritten

Lines 210, 214, …etc: please write in-vitro in italic form

For the discussion on the implication of oxidative stress on breast cancer cells death, please consider these papers: https://doi.org/10.3390/molecules27072108; https://doi.org/10.1055/a-0762-3775

It is preferably that the authors investigate the effect of the extracts on the antioxidant system through the measurement of specific markers, for example H2O2 and MDA levels and through enzyme activities for example catalase, peroxidase, superoxide dismutase, …etc.

It is advisable for the authors to improve the English of the manuscript before resubmission.

Reviewer 2 Report

The manuscript entitled "The Bioactive Properties of Carotenoids from Lipophilic Sea Buckthorn Extract (Hippophae rhamnoides L.) in Breast Cancer Cell Lines" is well-written, and contains relevant data that can be published after major revisions.

1. Abstract: If the authors have chosen to use the common name of the plant, they should not mix with the scientific name as observed at line 29. It creates confusion for the readers.

2. Methods:

The authors mentioned that "LSBE concentrations were calculated using a weighted average molecular mass of 557.75 g/mol, as resulted from HPLC analysis". Based on this statement, concentrations of LSBE tested in all experiments are expressed in µM. I don't think that this makes sense by expressing the concentrations of an extract (LSBE) in µM. The extract is a mixture of several compounds, and an explanation should be given regarding the scientific basis of calculating an average molecular weight of an extract.

Antioxidant assays: It will be better to test different concentrations of the extract and zeaxanthin in order to determine their IC50 values. Besides, in each experiment, a positive control should be included.

Cytotoxicity assay: No positive control was used.

General ROS method: The procedure used in this test is problematic. The cells should be treated with the extract or zeaxanthin first before induction of oxidative stress. I don't think that the exposure for 1h with these tested samples might be able to repair damage caused after pre-treatment with H2O2. But if the cells are pre-treated with the tested samples, they will incorporate the samples, and therefore be able to prevent oxidative stress.

The manuscript is well-written.

Reviewer 3 Report

The manuscript entitled „The Bioactive Properties of Carotenoids from Lipophilic Sea Buckthorn Extract (Hippophae rhamnoides L.) in Breast Cancer Cell Lines” presents interesting issue, but some problems should be corrected.

Abstract:

The aim of the study should be clearly formulated (e.g. “The aim of the study was…”)

The materials and methods should be briefly described

Introduction:

Authors present excessive number of information, including also a number of general and even trivial information that should not be presented in a scientific manuscript (e.g. “Even though more than 40 carotenoids are present in the human diet, only six of them: lycopene, α- and β-carotene, lutein, zeaxanthin, and β-cryptoxanthin, are present in the blood”, or “Elaeagnus rhamnoides (L.) A. Nelson (Hippophae rhamnoides) frequently known as sea buckthorn, is a deciduous shrub with yellow or orange berries, originally from China and it is found in the major temperate zones of the world, like France, Russia, Mongolia, India, Great Britain, Denmark, the Netherlands, Germany, Poland, Romania, Finland, and Norway.”) – Authors should be aware that they do not prepare the basic manual for students, or column of the newspaper, but a scientific paper that should be interesting for researchers from the area of food and nutritional sciences, so they should understand that their readers will have the nutritional knowledge.

Authors should focus on the really accurate information that are not too general – e.g. “Sea buckthorn berries are one of the most nutritious and vitamin-rich fruits produced by any plant as they are rich in antioxidants (tocopherols, carotenoids, flavonoids), fat- or water-soluble vitamins/provitamins (vitamins C and E, β-carotene), other carotenoids (lycopene and zeaxanthin), phytosterols, unsaturated fatty acids (especially omega-7 palmitoleic acid), amino acids, and minerals (iron, calcium, etc.)” – (1) is it really “one of the most nutritious and vitamin-rich fruits”? – if so, Authors should present comparison. (2) is it really source of iron and calcium? – if so, Authors should present specific amount and compare with the other products

The aim of the study should be clearly formulated (e.g. “The aim of the study was…”)

Results:

Instead of Figure 3, 4 and 6b, rather tables should be presented to be easier to follow

Figure 3 – n should be clearly indicated – it is described as n=3, but the figure presents n=12 for each concentration

The SEM is not properly applied, as it measures rather a precision for the estimated population mean and does not present the variability of data around the mean (while SD does), so instead of SEM, a SD should be applied.

After verifying the normality of distribution, distribution should be clearly defined and in case of parametric distribution mean ± SD should be presented, while for nonparametric distribution – median accompanied by minimum and maximum value.

Discussion:

The limitations of the study hold be extensively described (including the sample size, representativeness of the samples, etc.)

Materials and Methods:

Authors should present the materials for each assessment precisely – what was the number of samples and repetitions. The analysis conducted in triplicate may be either conducted 3 times for the same sample, or once for 3 different samples. It should be clearly indicated for each method applied.

“Plastic bags” should be defined (LDPE? PVC? Other?)

The SEM is not properly applied, as it measures rather a precision for the estimated population mean and does not present the variability of data around the mean (while SD does), so instead of SEM, a SD should be applied.

After verifying the normality of distribution, distribution should be clearly defined and in case of parametric distribution mean ± SD should be presented, while for nonparametric distribution – median accompanied by minimum and maximum value.

Conclusions:

Authors should not reproduce results but present more general statements

Authors’ Contributions:

Authors should properly define the contributions – e.g. what do Authors mean by “validation” if they did not conduct any validation in their study? There are similar doubts in case of “resources” and “data curation”

It seems that contribution of majority of authors (except for SV and OB) was only minor and none of them participated in preparing manuscript (but they only helped main Authors in methodology). There is a serious risk of a guest authorship procedure which is forbidden. In such case (if they did not participate in manuscript preparation in any way) they should be rather presented in Acknowledgements Section and not be indicated as authors of the study.

Reviewer 4 Report

Given the large number of analyzed data, this is an interesting study with a possible impact in this area. 

Introduction. In such a vast field, please use recent, relevant references. Please revise accordingly.

Please sharpen the description of the novelty factor in the "in this work" section of the introduction. What exactly was done in this study for the first time? It is important to put this article in perspective to enable readers to quickly decide whether to read the paper or not.

Results. Please improve the quality of Figures 1 and 2 by using an appropriate software (ex. Origin). Figure 2. Please keep only the positive value on the axis. Please improve the quality of Figures 2-4, i.e. the font and size of letters.

Conclusions section should be short with important observations. Please re-write to be a paragraph that describes what is meant to be taken away from reading the study (also values), not a summary of results as listed here.

 Moderate editing of English language.

Round 2

Reviewer 1 Report

The authors answered the questions that have been raised and I have no more comments

No comment

Author Response

We appreciate your precious time and effort in reviewing our paper and providing valuable comments that helped to improve the current version of the manuscript. The authors have carefully considered the comments, tried their best to address every one of them, and would like to express gratitude to the reviewer for the positive feedback addressed to our manuscript.

Reviewer 2 Report

The explanation about expressing the concentrations of LSBE in µM seems to be convincing. But, I will suggest to expressed the LSBE concentrations in both µM and µg/mL, and give an explanation for the conversion of LSBE concentrations into µM that will be available for the readers of this manuscript.

Tables 4 and 5: why is the concentration expressed in μM/ml?

Reviewer 3 Report

The manuscript entitled „The Bioactive Properties of Carotenoids from Lipophilic Sea Buckthorn Extract (Hippophae rhamnoides L.) in Breast Cancer Cell Lines” presents interesting issue, but some problems should be corrected.

Discussion:

The limitations of the study should be extensively described (including the sample size, representativeness of the samples, etc.)

Materials and Methods:

Authors should present the materials for each assessment precisely – what was the number of samples and repetitions. The analysis conducted in triplicate may be either conducted 3 times for the same sample, or once for 3 different samples. It should be clearly indicated for each method applied.

Line 397 – “bags bags”

Conclusions:

Authors should neither reproduce results, nor reproduce Abstract, but should present more general statements

Authors’ Contributions:

It seems that contribution of majority of authors (except for SV and OB) was only minor and none of them participated in preparing manuscript (but they only helped main Authors in methodology). There is a serious risk of a guest authorship procedure which is forbidden. In such case (if they did not participate in manuscript preparation in any way) they should be rather presented in Acknowledgements Section and not be indicated as authors of the study.

Reviewer 4 Report

The manuscript can be accepted for publication.

Author Response

(The authors gave the same response as above.)
